# Review of the Synthesis and Anticancer Properties of Pyrazolo[4,3-*e*][1,2,4]triazine Derivatives

**DOI:** 10.3390/molecules25173948

**Published:** 2020-08-29

**Authors:** Zofia Bernat, Anna Szymanowska, Mateusz Kciuk, Katarzyna Kotwica-Mojzych, Mariusz Mojzych

**Affiliations:** 1Department of Chemistry, Siedlce University of Natural Sciences and Humanities, 3 Maja 54, 08-110 Siedlce, Poland; zosiabernat@wp.pl; 2Department of Biotechnology, Medical University of Bialystok, Kilinskiego 1, 15-222 Bialystok, Poland; anna.szymanowska@umb.edu.pl; 3Department of Molecular Biotechnology and Genetics, Laboratory of Cytogenetics, University of Lodz, Banacha 12/16, 90-237 Lodz, Poland; mateusz.kciuk@unilodz.eu; 4Doctoral School of Exact and Natural Sciences, University of Lodz, Banacha Street 12/16, 90-237 Lodz, Poland; 5Department of Histology, Embryology and Cytophysiology, Medical University of Lublin, Radziwiłłowska 11, (Collegium Medicum), 20-080 Lublin, Poland; katarzynakotwicamojzych@umlub.pl

**Keywords:** pyrazolo[4,3-*e*][1,2,4]triazines, anticancer activity, sulfonamides, 1,2,4-triazine, fused tetrazole derivatives, fluviols

## Abstract

This review focuses on the cytotoxic effect of new synthetic pyrazolo[4,3-*e*][1,2,4]triazine derivatives against different tumor cell lines. Some annulated pyrazolotriazines i.e., pyrazolo[4,3-*e*][1,2,4]triazolo[4,3-*b*][1,2,4]triazines and pyrazolo[4,3-*e*]tetrazolo[1,5-*b*][1,2,4]triazine demonstrated significant broad cytotoxic activity in micromolar range concentration, which could have excellent potential to be new candidate therapeutic agents in cancer chemotherapy.

## 1. Introduction

Despite the significant progress that has been made in recent years in the treatment of cancer, tumors still represent a high risk for humanity, and therefore the development of effective cancer therapy is still a challenge for modern medicine. The choice of treatment depends on the type of cancer and its stage at the time of diagnosis. Both chemotherapy, radiotherapy, and immunotherapy have a number of side effects that reduce the effectiveness of drugs and can reduce the quality of life of patients. Therefore, it is extremely important to search for new therapeutic strategies, as well as to develop drugs with a better pharmacological profile that act specifically on cancer cells and do not have a toxic effect on normal cells. The search for new lead structures involves identifying new chemical compounds that affect well-defined molecular targets. Triazine derivatives are an interesting group of compounds with potential anti-tumor activity, and this scaffold has been used in oncological therapy since 1965. 

Since that time, a number of studies have been undertaken to modify the structure of 1,2,4-triazine derivatives, which were to provide molecules with stronger cytotoxic properties and at the same time giving fewer side effects. Thanks to this research, a number of benzo- or hetero-fused 1,2,4-triazine derivatives have been found and described in the literature as new antitumor agents. Compounds in which the 1,2,4-triazine nucleus is condensed with five-membered heterocycles have received considerable attention because they are bioisosteric with purine core. Among the 1,2,4-triazine condensed with one heterocycle, compounds bearing a pyrrole ring, such as pyrrolo[2,1-*c*][1,2,4]triazine and pyrrolo[2,1-*f*][1,2,4]triazine, represent the most abundant class of triazine with antitumor activity. 

However, another very interesting group of fused 1,2,4-triazines with five-membered heterocycles are derivatives of the little-known pyrazolo[4,3-*e*][1,2,4]triazine ring system. It is a novel scaffold which, despite of the fact that it is an important source for bioactive molecules, has been less studied so far in comparison with the pyrrolotriazines. In the past few decades, the isolation and structural characterization of seven naturally occurring pyrazolo[4,3-*e*][1,2,4]triazines: pseudoiodinine [1], nostocine A [2], and fluviols A–E [3] (Figure 1) have been reported. These natural compounds with wide antibiotic and antitumor activities were found as extracellular metabolites of some microorganism of the class *Pseudomonas fluorescens* var. *pseudoiodinine* and *Nostoc spongiaeforme*. Structures of two natural pigments of this group, namely nostocine A and fluviol A (*normethylpseudoiodinine*), have been clearly defined by X-ray crystallographic analysis [2] and further confirmed by total synthesis [4]. The data cited above indicate the role of this heterocyclic system in the search for new pharmacologically active compounds.

This review presents the methods for the synthesis and functionalization of the pyrazolo[4,3-*e*][1,2,4]triazine ring system, which in the group of fused pyrazolotriazines is little known and also little described in the literature. It is known that proper functionalization of the heterocyclic core is a key element to design new molecules with potential biological properties. Therefore, the collection of the results of the current scientific research on this valuable heterocyclic system is necessary and justified. The results of experimental studies published so far have shown that a series of derivatives of this heterocyclic system possess various biological activity including antitumor property, which is the main and leading topic of this review. The paper also presents the methods used for the preparation of pyrazolo[4,3-*e*][1,2,4]triazine sulfonamide derivatives, their activity against cancer cell lines, and the inhibition of two carbonic anhydrase isozymes (CAIX and CAXII) that are highly overexpressed in hypoxic tumors and show very restricted expression in normal tissues. Among the presented data, the tricyclic pyrazolo[4,3-*e*][1,2,4[triazines fused with triazole or tetrazole ring are of particular interest. They constitute new groups of heterocyclic systems and are characterized by high antitumor activity, and it seems that they may be a source of new chemotherapeutic agents.

## 2. Approach to the Construction of Pyrazolo[4,3-*e*][1,2,4]triazine Ring System

### 2.1. Synthesis from 4-Amino-5-acylhydrazinopyrazole

There are few different methods described in the literature for the construction of 1,3,5-trisubstituted pyrazolo[4,3-*e*][1,2,4]triazines [4,5,6,7,8,9,10,11,12]. These methods can be divided into two groups, one including the building of the 1,2,4-triazine core on a pyrazole derivative (Scheme 1) and the second one incorporating the construction of the pyrazole ring onto the 1,2,4-triazine nucleus [4,5,6,12] (Scheme 2) [10,11].

### 2.2. Synthesis from 3,5-Disubstituted-1,2,4-triazin-6(1H)-one with Cytotoxic Activity

In a similar manner 1,5-diaryl-3-(3,4,5-trimethoxyphenyl)pyrazolo[4,3-*e*][1,2,4]triazines have been synthetized by Gucký et al. (Scheme 3) [12]. These derivatives showed selective inhibitory activity against the A549 cell line, while they are generally less active against leukemia cell lines, including otherwise highly chemosensitive CEM lymphoblasts. The results of cytotoxic activity for some derivatives are summarized in Table 1.

The above-mentioned methods for the synthesis of the pyrazolo[4,3-*e*][1,2,4]triazine derivatives had a little importance to further study on their functionalization and determining structure-activity relationship (SAR) because they did not contain appropriate functional groups.

### 2.3. Synthesis from 3-Substituted-1,2,4-triazine with Cytotoxic Activity

This fact encouraged Mojzych’s research group to develop a new, simple and useful way for the synthesis of the mentioned bicyclic system from readily available oximes of 5-acyl-1,2,4-triazines [13] or ketones [14] with hydrazine derivatives. The new methods are based on nucleophilic substitution reaction of hydrogen and allows for obtaining different pyrazolo[4,3-*e*][1,2,4]triazines useful to determine their structure-activity relationship [8,9,15,16]. 5-Acyl-1,2,4-triazine derivatives were obtained in good yield in the reaction of 1,2,4-triazines with nitroalkane anions under nucleophilic substitution of hydrogen (Scheme 4) [13]. In this process, nitroalkanes play a role of masked acylating agent yielding in the first step oximes of 5-acyl-1,2,4-triazines, which may be easily converted to the corresponding ketones [14] which were used to receive the corresponding hydrazones as a key intermediate for the construction of 1H-pyrazolo[4,3-*e*][1,2,4]triazines (Scheme 4) [8]. The method seems to be general and tolerates a wide range of substituents and, comparing this method with the previously described in literature, we can easily see that the approach does not need a good leaving group in the position C6 of the triazine ring. It has to be noted that pyrazolo[4,3-*e*][1,2,4]triazine formation strongly depends on the kind of substituent on the phenyl ring of phenylhydrazones. The shorter reaction time and higher yield was observed for electron donating groups (methyl or methoxy group), but, for electron withdrawing groups (Cl, NO_2_), the time reaction was much longer and the yield was low, which could be observed for the NO_2_ group. Based on these observations, we have proposed the mechanism of the intramolecular ring closure of hydrazones [8,16]. The most probable reaction proceeds via the protonated hydrazone intermediate, followed by ring closure involving intramolecular electron pair attack of a hydrazine nitrogen atom onto the C6 of 1,2,4-triazine ring to give the adduct σ^H^, that, via an air oxidation, gives final pyrazolo[4,3-*e*][1,2,4]triazine (Scheme 4).

We have to underline that the new method for the preparation of pyrazolo[4,3-*e*][1,2,4]triazine derivatives is convenient to introduce a number of substituents at N1 and C3 position of the pyrazole ring. Introduction of new substituents at the C5 position is possible by nucleophilic substitution reaction of a suitable nucleofugal group. However, the methylsulfanyl group in pyrazolo[4,3-*e*][1,2,4]triazine was unreactive in the reaction with nucleophilic agents. In order to increase the activity of the heterocyclic system in the S_N_Ar reactions, the methylsulfanyl group was replaced with more electron-withdraw methylsulfonyl substituent. This group is considered as one of the better leaving groups in nucleophilic substitution reactions of 1,2,4-triazine [17,18]. Utilizing the methylsulfonyl group, Mojzych et al. developed a method for the functionalization of pyrazolo[4,3-*e*][1,2,4]triazine system via nucleophilic substitution at the C5 position using O-, N-, S- and C-nucleophiles (Scheme 5) [19].

Using the nucleophilic substitution reaction of the methylsulfonyl group at the C5 position of 1H-pyrazolo[4,3-*e*][1,2,4]triazine, a series of new derivatives of this system were obtained.

In the group of simple synthesized pyrazolo[4,3-*e*][1,2,4]triazines [8,9,15,16,19,20,21], only a few derivatives showed moderate activity against variety of human tumor cell lines: prostate cancer (PC-3), breast cancer (MCF-7), non-small-cell lung cancer (H460), colorectal adenocarcinoma (Colo205). Their structures and cytotoxicity are presented in Table 2. It is worth noting that the 3-methyl-1-phenyl-5-phenylaminopyrazolo[4,3-*e*][1,2,4]triazine derivative showed the highest degree of reduced cell viability with IC_50_ value 4 µM in Colo205 cells.

## 3. Synthesis and Anticancer Activity of Pyrazolo[4,3-*e*][1,2,4]triazine Sulfonamides

### 3.1. Synthesis from 5-Methylsulfanylpyrazolo[4,3-e][1,2,4]triazine

The lack of significant antitumor activity in the group of simple substituted pyrazolotriazine derivatives encouraged scientists to complete further functionalization of the heterocyclic core. The combination of the naturally occurring pyrazolo[4,3-*e*][1,2,4]triazine ring system with pharmacophore groups enabled the design of new derivatives with higher potential biological activity.

One of the most important pharmacophore groups is a sulfonamide moiety characteristic for many chemical compounds used in medicine [22,23]. Their importance stems from the fact of diverse biological activity which includes antibacterial, antimalarial, hypotensive, diuretic, hypoglycemic, antithyroid, antiparasitic, anti-inflammatory, and antiglaucomatous properties [24]. Furthermore, research studies have shown that sulfonamides may exhibit an antitumor effect by inhibiting the activity of protein kinases including cyclin-dependent kinases (CDKs) [25,26] or carbonic anhydrase (**CA**; EC 4.2.1.1) [27,28,29].

Protein kinases participate in many signal transduction pathways including those involved in growth, differentiation, and cell division. The overexpression or mutation of some protein kinases can lead to cancer. Several protein kinases represent targets for cancer chemotherapy. These targets include the Bcr-Abl protein kinase, the RAF protein-serine/threonine protein kinase, the epidermal growth factor receptor protein tyrosine kinase, protein kinase C, and anaplastic lymphoma protein-tyrosine kinase [30,31,32]. In chronic myelogenous leukemia, the reciprocal translocation between chromosomes 9 and 22 lead to the chimeric formation of a portion of the Bcr gene and the Abl gene. The product of this translocation is Bcr-Abl p210 protein isoform with tyrosine kinase activity. The Abl gene was first described in the genome of the Abelson murine leukaemia virus. The Bcr-Abl oncoprotein was a target for drug discovery, and imatinib (STI 571, Gleevec) was one product of this research. Gleevec, an ATP analog, is a specific and competitive inhibitor of the Bcr-Abl protein kinase that is being used to treat chronic myelogenous leukemia.

The cyclin-dependent kinases (CDKs) are a family of Ser/Thr kinases, which, in association with specific cyclins, play critical roles as regulators of the different phases of the cell cycle. These enzymes and their direct regulators are frequently mutated, amplified, or deleted in malignant cells, suggesting that pharmacological CDK inhibition may be an effective strategy for treating cancer [33].

During the last decade, carbonic anhydrase became an attractive and promising scientific target for anticancer therapy since two cancer-associated isozymes CA IX and XII [29,34,35,36,37,38,39,40] were found to be overexpressed in many tumors [41,42]. These two transmembrane proteins play a key role in tumor progression and response to treatment [34]. It has been demonstrated that CA IX is overexpressed in hypoxic tumor, participate in acidification of the environment of tumor cells, and contribute to disease progression giving a poor prognosis for treatment. As CA IX is an important oncotarget, much attention has been focused to find new CA IX inhibitors as anticancer drugs.

The first sulfonamide derivatives of the pyrazolo[4,3-*e*][1,2,4]triazine were prepared according to Scheme 6 and their antitumor activity was tested. One group constitutes 5-phenylaminosulfonamide derivatives of pyrazolo[4,3-*e*][1,2,4]triazine **9a**–**l** [43] being analogs of known inhibitors of protein kinases and the other group includes sildenafil analogs **12a**–**l** in which HN-C=O moiety has been replaced by two triazine nitrogen atoms [44,45].

Due to the similarity of derivatives **9a**–**l** to known inhibitors of protein kinases, only this group of sulfonamides has been studied as inhibitors of the Abl protein kinase. In tests, the most active compounds were **9c** and **9e**. Their IC_50_ values are expressed in micromolar concentration range (IC_50_ = 5.8–5.9 µM). To better understand the activity of pyrazolo[4,3-*e*][1,2,4]triazines **9c** and **9e** and the binding of Abl, kinase molecular modeling was performed, the results of which suggested that compounds **9c** and **9e** might bind to Abl in a similar manner as described for the pyrido[2,3-*d*]pyrimidine PD180970, interacting with the protein via non-polar interactions and hydrogen bonds with the NH group of the amino acid M318 in the main chain [46].

On the other hand, this sulfonamide group displayed lack of activity towards CDK2. Molecular docking suggested that the negative results of the biochemical assays are due to the relatively unfavorable mode of binding adopted by the pyrazolo[4,3-*e*][1,2,4]triazines in the CDK2 active site [43].

The sulfonamides **9a**–**l** were also investigated against leukemia cell lines (K562, BV173, HL60, CCRF-CEM) using the MTT assay (Table 3) [43,44]. The concentration-dependent activity was observed for all tested compounds **9a**–**l** and **12a**–**l**. The breast carcinoma cell lines were much less sensitive to the tested compounds in comparison to the leukemia cell lines. It is noteworthy that the IC_50_ values for the most active derivatives **9e** and **9c** against leukemia cells are 5–7 times lower than the IC_50_ values for the breast cancer cells. This fact suggests that the tested compounds exhibit significant selectivity for tumor cells.

Moreover, for both sulfonamide groups, the potential anticancer activity in MCF-7 and MDA-MB-231 cells was determined by [^3^H]thymidine incorporation assay and MTT test, where proliferation and viability of breast cancer cells were analyzed (Table 4 and Table 5) [44]. All tested compounds showed concentration dependent activity but with different potency.

The influence of sulfonamides **9a–k** on collagen biosynthesis in breast cancer cells (MCF-7 and MDA-MB-231) was also examined (Table 6). In both cell lines, compound **9a** was found to be more effective inhibitor of collagen biosynthesis than chlorambucil. IC_50_ for **9a** and chlorambucil (in MDA-MB-231: 47 µM and 52 µM, in MCF-7: 58 µM and 72 µM, respectively) showed specific inhibitory effect of compound **9a** on collagen biosynthesis.

Biological research revealed that both classes of sulfonamide derivatives (**9a**–**l** and **12a**–**l**) had cytotoxic activity against estrogen receptor positive breast cancer cells—MCF-7 and estrogen receptor negative breast cancer—MDA-MB-231. In addition to this, compounds **9a**–**l** affect collagen synthesis, which may have a role in metabolism and function of human breast cancer cells.

Obtained sulfonamides **9a**–**l** were also evaluated for their inhibitory potency against carbonic anhydrase, particularly against two isozymes, namely cancer-associated isoforms hCA IX and XII [44]. The best results against hCA IX were observed for sulfonamide **9h** (K_I_ = 23.7 nM) and **9d** (K_I_ = 26.5 nM), which were similar to results obtained for the standard—acetazolamide (K_I_ = 25 nM) (Table 6). The best activity was observed in tests against hCA XII. In this study, all derivatives showed a good inhibition of the enzyme with K_I_ in the range of 5.3 nM to 9.0 nM. The lowest value of K_I_ was observed for derivative **9a** (K_I_ = 5.3 nM), which is the best chemotherapeutic agent among all investigated sulfonamides. The tumor-associated isoforms hCA IX and XII were inhibited by some of the investigated derivatives. Thus, hCA IX was not at all inhibited by four of the new derivatives (**9a**, **9c**, **9e** and **9g**), was weakly inhibited by two of them (**9f** and **9k**), whereas **5**, **6**, **9d**, **9h** and **9i** were more effective as hCA IX inhibitors, with K_I_s in the range of 23.7–89 nM. On the contrary, hCA XII was potently inhibited (K_I_s < 10 nM) by all the reported compounds (Table 7).

### 3.2. Synthesis from 3-Methylsulfanyl-1,2,4-triazine

Continuing research on sildenafil analogs a new approach to the synthesis of the sulfonamides was reported and their cytotoxicity against two human cancer cell lines: breast cancer (MCF-7) and human myelogenous leukemia (K562) were determined (Scheme 7) [45].

The results of biological study showed that none of the sildenafil analogs exhibited cytotoxicity in the tested concentrations. In addition, the ability of the derivatives to inhibit protein kinase CDK2/cyclin E and Abl was investigated. In the group of tested compounds, only sulfonamide **12e** showed moderate activity against kinase CDK2 (IC_50_ = 44.3 µM). Other compounds are inactive.

However, the group of sulfonamides appeared to be active against tumor associated carbonic anhydrase hCA IX and hCA XII [45]. The most active inhibitors against hCA IX were derivatives **12p** (K_I_ = 15.4 nM) and **12b** (K_I_ = 24.4 nM). Compounds **12r** (K_I_ = 3.8 nM) and **12i** (K_I_ = 5.5 nM) are the most active structures against hCA XII. The other compounds showed activity against hCA XII with K_I_ value in the range of 40–610 nM. Moreover, the structure of the sildenafil analogs was confirmed by X-ray analysis performed for the single crystal of derivative **12i** [45].

Another group of sulfonamides constitutes sildenafil analogs in which the methyl group at the nitrogen atom N1 on the pyrazole was replaced by the aryl ring (Scheme 8) [47]. The aim of this study was to investigate whether the replacement of a methyl group by aryl substituent will have an effect on the antitumor activity and inhibition of human carbonic anhydrase isozymes.

Obtained sulfonamides were subjected to biological tests against breast carcinoma cells MCF-7 and MDA-MB-231 [47]. The most active derivatives are compounds **24b**, **24d** and **24i**, which showed moderate cytostatic activity against MCF-7 and MDA-MB-231 cells with IC_50_ value in the range of 126 ± 2 µM–185 ± 2 µM (Table 8). Other derivatives were inactive. In order to verify the mechanism responsible for the growth inhibitory effect on cancer cells, biosynthesis DNA in the presence of sulfonamides **24a–k** and chlorambucil as a standard was examined. The concentration of **24b**, **24d,** and **24i** necessary to inhibit the biosynthesis DNA in human breast cancer cells MCF-7 and MDA-MB-231 by 50% (IC_50_) was in the range of 132 ± 2 µM do 173 ± 2 µM. For the other compounds, necessary concentration to inhibit [^3^H]thymidine incorporation into DNA by 50% was found to be more than 200 µM.

Despite moderate cytostatic activity, sulfonamides achieved have shown greater inhibition of human carbonic anhydrase isoenzymes. hCA IX was efficiently inhibited by most of the obtained compounds, with inhibition constants ranging between 13.8 and 417 nM (Table 9) [47]. Poor inhibition of this isoforms (K_I_s of 403–417 nM) showed derivatives **24i** and **24j**. Furthermore, these studies demonstrated that both the primary sulfonamides and the tertiary ones showed similar inhibition, although their mechanisms of inhibitory activity are very different. The primary sulfonamide binds to the metal ion, whereas the tertiary ones probably in the coumarin-binding site. hCA XII was also inhibited by the new reported compounds with inhibition constants ranging between 70.3 and 536 nM. Compound **24a** was a poor hCA XII inhibitor (K_I_ of 536 nM), whereas the remaining ones were medium potency inhibitors with K_I_s in the range of 70.3–93.1 nM.

## 4. Synthesis and Anticancer Activity of Annulated Pyrazolo[4,3-*e*][1,2,4]triazines: Pyrazolo[4,3-*e*][1,2,4]triazolo[4,3-*b*][1,2,4]triazines and Pyrazolo[4,3-*e*]tetrazolo[1,5-*b*][1,2,4]triazines

A very interesting group of compounds with anticancer activity constitute the fused pyrazolo[4,3-*e*][1,2,4]triazines with 1,2,4-triazole or tetrazole ring. The general synthesis pathway leading to the tricyclic derivatives is depicted in Scheme 9 [21,48,49].

Preliminary biological studies have shown that the obtained tricyclic derivatives have anti-cancer properties, which were demonstrated in tests on various tumor cell lines. The highest antitumor activity (in the nanomolar concentration range) was shown by tetrazole derivative **30b** (Table 10) [21]. At this point, it should be added that, in heterocyclic systems with a terminal tetrazole ring, a valence tautomerism may occur i.e., the equilibrium between the azide derivative and the tricyclic system [48,50,51]. The shift of the equilibrium depends on the environment of this process, e.g., solvent properties [48,50]. Tautomeric equilibrium is an important and interesting chemical phenomenon because the various tautomers of the same compound have different physico-chemical properties. Therefore, the same compound may have different reactivity, or even biochemical properties, depending on the tautomeric form. For that reason, prediction of the tautomeric mixture composition is important for the design of new biologically active compounds as well as technological processes or understanding processes of life. Thus, the research results on tautomeric equilibrium in pyrazolo[4,3-*e*][1,2,4]triazine derivatives fused with tetrazole ring were described in literature [48,50].

Another valuable group of tricyclic compounds with anti-cancer properties are pyrazolo[4,3-*e*][1,2,4]triazolo[4,3-b][1,2,4]triazine derivatives. Their antiproliferative activity was evaluated against human lung cancer A549 and colon cancer LS180 using MTT tests. Obtained results revealed a concentration-dependent decrease in cancer cell proliferation. It was observed that pyrazolo-triazolo-triazines were more active than the intermediates. As shown in Table 11, compound **28c** with chloromethyl substituent was the most active. In contrast, compound **28b** with methyl substituent was the least active. It was observed that lung cancer A549 cells were more sensitive for pyrazolo[4,3-*e*][1,2,4]triazolo[4,3-b][1,2,4]triazine derivatives action than colon cancer LS180 cells. It should be noted that the tested compounds showed higher antiproliferative activity than common cytotoxic drugs, cisplatin (lung carcinoma), and 5-fluorouracil (colon adenocarcinoma).

## 5. Conclusions

In the review, we have summarized the results of research on the synthesis and anticancer activity of the new synthetic derivatives of the little-known pyrazolo[4,3-*e*][1,2,4]triazine ring system. One of the presented derivatives constitute sulfonamides with potential antitumor activity on the cancer cell lines MCF-7, MDA-MB-231, BV173, HL60, CCRF-CEM, and the ability to inhibit protein kinases Bcr-Abl and CDKs, as well as two isozymes of carbonic anhydrase. Positive results were obtained for inhibition of isoforms hCA IX and hCA XII associated with cancer. Derivatives **9d** and **9h** are the most active hCA IX inhibitors, whereas compounds **9a**, **9c** and **9e** show the highest activity against hCA XII and are also the best cytostatics among all investigated sulfonamides. Another very interesting group with good cytostatic activity were derivatives of annulated pyrazolo[4,3-*e*][1,2,4]triazines i.e., pyrazolo[4,3-*e*][1,2,4]triazolo[4,3-*b*][1,2,4]triazines and pyrazolo[4,3-*e*]tetrazolo[1,5-*b*][1,2,4]triazine. Presented compounds showed higher antiproliferative activity than popular cytostatics such as cisplatin (lung carcinoma) and 5-fluorouracil (colon adenocarcinoma). Therefore, they may constitute a new group of candidates for drugs useful in the treatment of various cancers.

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
