# Peer review of "Review of the Synthesis and Anticancer Properties of Pyrazolo[4,3-e][1,2,4]triazine Derivatives"

_molecules, 2020, doi:10.3390/molecules25173948_

Round 1
Reviewer 1 Report
Comments and Suggestions for Authors:
The manuscript exhibits several weaknesses that should be corrected and re-edited.
Manuscript sections: It is suggested as the following.
1. Introduction (Figure 1.)
- Approach to the construction of pyrazolo[4,3-e][1,2,4]triazine ring system
2.1. Synthesis from 4-amino-5-acylhydrazinopyrazole or 4-amino-5-aroylhydrazinopyrazole (Scheme 1.)
2.2. Synthesis from 3,5-disubstituted-1,2,4-triazin-6(1H)-one with cytotoxic activity (Scheme 2., Scheme 3.; Table 1.)
2.3. Synthesis from 3-substituted-1,2,4-triazine with cytotoxic activity
(Scheme 4., Scheme 5.; Table 2.)
- Synthesis and anticancer activity of pyrazolo[4,3-e][1,2,4]triazine sulfonamides
3.1. Synthesis from 5-methythiopyrazolo[4,3-e][1,2,4]triazine
(Scheme 6.; Figure 2., Figure 3.; Table 3.,Table 4., Table 5. (original Table 6.), Table 6. (original Table 7.))
3.2. Synthesis sildenafil analogs of pyrazolo[4,3-e][1,2,4]triazine sulfonamides
3.2.1. Synthesis from 5-methylthiopyrazolo[4,3-e][1,2,4]triazine
(Scheme 7. (original Scheme 6.); Table 7. (original Table 5.))
3.2.2. Synthesis from 3-methylthio-1,2,4-triazine
(Scheme 8. (original Scheme 7.), Scheme 9. (original Scheme 8.); Table 8., Table 9.)
- Synthesis and anticancer activity of annulated pyrazolo[4,3-e][1,2,4]triazines: pyrazolo[4,3-e][1,2,4]triazolo[4,3-b][1,2,4]triazines and pyrazolo[4,3-e]tetrazolo[1,5-b][1,2,4]triazines
(Scheme 10. (original Scheme 9.); Table 10., Table 11.)
- Summary and Conclusions
There are some notes, as follows:
- Lines 22, 359, 365, 369, 382 and 383: The nomenclature of pyrazolo[4,3-e]triazolo[4,5-b][1,2,4]triazines should be corrected as pyrazolo[4,3-e][1,2,4]triazolo[4,3-b][1,2,4]triazines and pyrazolo[4,3-e]tetrazolo[4,5-b][1,2,4]triazine should be corrected as pyrazolo[4,3-e]tetrazolo[1,5-b][1,2,4]triazines. Triazole have two types 1,2,3-triazole and 1,2,4-triazole.
- Line 70: In Scheme 1. give the full name of PPA in Abbreviations
- Lines 76, 82 and 85: The nomenclature of 3,7-diaryl-5-(3,4,5-trimethoxyphenyl)-pyrazolo[4,3-e][1,2,4]triazines should be corrected as 1,5-diaryl-3-(3,4,5-trimethoxyphenyl)-pyrazolo[4,3-e][1,2,4]triazines.
- Line 80: In Scheme 3. the first structure loss one hydrogen at N-1 of 1,2,4-triazine ring.
- Line 120: The “electronegative” should be corrected as “electron-withdraw”.
- Line 135: In Table 2. at least this compound 1-phenyl-3-methyl-5-aminophenylpyrazolo[4,3-e][1,2,4]triazine should be mentioned in text.
- Line 174: The “N1,N4-disudstituted” should be corrected as “5-phenylamino”.
- Line 176: The structure of sildenafil should be given in the text.
- Line 252: Compounds 9h and 9d should be corrected as 9g and 9i.
- Line 259: Compounds 9a, 9c, 9e and 9g should be corrected as 9a, 9b, 9d and 9f.
- Line 259: Compounds 9f and 9k should be corrected as 9e and 9h.
- Lines 259-260: Compounds 5, 6, 9h, 9d and 9i should be corrected as 5, 6, 9g, 9i and 9j.
- Line 291: In scheme 8. all atoms font should be given in the Arial type.
- Lines 378-379: Compounds 9d and 9h should be corrected as 9g and 9i.
- Lines 379: Compounds 9a, 9c and 9e should be corrected as 9a, 9b and 9f.
- Line 331: The “with triazole” should be corrected as “with 1,2,4-triazole”.
- Other minor mistakes in lines 216, 220, 268, 394, 403, 407, 413, 425, 446 (it is not a complete list).

Author Response
The responses to the comments are listed below point by point.
- “Manuscript sections: It is suggested as the following.”
We have restructured the revised manuscript, and added sub-headings suggested by the Reviewer. Although dividing section number 3: “The Synthesis and anticancer activity of pyrazolo[4,3-e][1,2,4]triazine sulphonamides” into 4 smaller sections is not necessarily. Presenting both groups of compounds 9a-l and 12a-l in one paragraph, helps to compare the effect of the -NH group on the biological activity of both very similar sulfonamides on cancer cell lines and understand the molecular mechanism of two similar classes of compounds. According to the Reviewer suggestion we divided this paragraph to two sections: 3.1 Synthesis from 5-methylsulfanylpyrazolo[4,3-e][1,2,4]triazine and 3.2 Synthesis from 3-methylsulfanyl-1,2,4-triazine.
- Lines 22, 359, 365, 369, 382 and 383: The nomenclature of pyrazolo[4,3-e]triazolo[4,5-b][1,2,4]triazines should be corrected as pyrazolo[4,3-e][1,2,4]triazolo[4,3-b][1,2,4]triazines and pyrazolo[4,3-e]tetrazolo[4,5-b][1,2,4]triazine should be corrected as pyrazolo[4,3-e]tetrazolo[1,5-b][1,2,4]triazines. Triazole have two types 1,2,3-triazole and 1,2,4-triazole.
According to the suggestion, we have checked the nomenclature of pyrazolo[4,3‑e][1,2,4]triazolo[4,5-b][1,2,4]triazines and pyrazolo[4,3-e]tetrazolo[4,5-b][1,2,4]triazine. The chosen pathern of numbering of each atom is presented below.
|
pyrazolo[4,3-e]tetrazolo[4,5-b][1,2,4]triazine |
pyrazolo[4,3‑e][1,2,4]triazolo[4,5-b][1,2,4]triazines |
- Line 70: In Scheme 1. give the full name of PPA in Abbreviations
We have clarified the definition of PPA in the Abbreviations section and added missing abbreviations.
- Lines 76, 82 and 85: The nomenclature of 3,7-diaryl-5-(3,4,5-trimethoxyphenyl)-pyrazolo[4,3-e][1,2,4]triazines should be corrected as 1,5-diaryl-3-(3,4,5-trimethoxyphenyl)-pyrazolo[4,3-e][1,2,4]triazines
The Reviewer’s comment is correct, and we have completed the structure with the lost hydrogen at N1 of 1,2,4 –triazine ring.
- Line 120: The “electronegative” should be corrected as “electron-withdraw”.
As suggested by the Reviewer, we have corrected the “electronegative” as “electron-withdraw”.
- Line 135: In Table 2. at least this compound 1-phenyl-3-methyl-5-aminophenylpyrazolo[4,3-e][1,2,4]triazine should be mentioned in text.
According to the Reviewer suggestion the most active compound against colorectal adenocarcinoma (Colo205) was mentioned in the text.
- Line 174: The “N1,N4-disudstituted” should be corrected as “5-phenylamino”.
We have corrected the “N1,N4-disudstituted” as “5-phenylamino” according to the Reviewer suggestion.
- Line 176: The structure of sildenafil should be given in the text.
- According to the suggestion the structure of sildenafil is given in the Scheme 6 to show the difference in structure of sildenafil and our new sildenafil analogues.
- Line 252: Compounds 9h and 9d should be corrected as 9g and 9i.
Line 259: Compounds 9a, 9c, 9e and 9g should be corrected as 9a, 9b, 9d and 9f.
Line 259: Compounds 9f and 9k should be corrected as 9e and 9h.
Lines 259-260: Compounds 5, 6, 9h, 9d and 9i should be corrected as 5, 6, 9g, 9i and 9j.
Lines 378-379: Compounds 9d and 9h should be corrected as 9g and 9i
Lines 379: Compounds 9a, 9c and 9e should be corrected as 9a, 9b and 9f.
We thank the Reviewer for pointing out our mistakes. We apologise for the confusing marks of the compounds. It was caused by incorrect values in the Table 7 . We have corrected the KI values.
- Line 291: In scheme 8. all atoms font should be given in the Arial type.
According to the suggestion, the font was unified in the Scheme 8.
- Line 331: The “with triazole” should be corrected as “with 1,2,4-triazole”.
According to the Reviewer's suggestion the “with triazole” was corrected as “with 1,2,4-triazole”
- Other minor mistakes in lines 216, 220, 268, 394, 403, 407, 413, 425, 446 (it is not a complete list).
Several errors were corrected. However the title in line 394 (Now 418-419) “Pyrazolo(4.3-e)as-triazine, a new heterocyclic system from Pseudomonas Fluorescens var. Pseudoidinum” is consistent with the original article published at Chemische Berichte.
We thank the Reviewer thoughtful and supportive comments. We have revised our manuscript in the response to Reviewer suggestions and believe that this improved manuscript is acceptable for publication in Molecules.

Reviewer 2 Report
The review of Mojzych and co-workers is an interesting overview on synthesis and anticancer activity of that particular class of organic derivatives represented by pyrazolo[4,3-e][1,2,4]triazines. Authors deal with the subject in a competent and comprehensive way. The synthetic protocols are well described and the data of biological activity of compounds have been selected in an appropriate manner. The quality of the schemes is good and the bibliographical references appear complete.
For all these reasons I have nothing against the publication of the review in Molecules,
My only observation is that authors have inadvertently exchanged scheme 1 with scheme 2 and vice versa.
Author Response
Thank you for taking the time to review our manuscript and for the very nice comment.
Reviewer 3 Report
The manuscript cannot be accepted for publication in the journal of Molecules since the tables and the figures are copied by other already published articles. For example
a. the figures 2-3 and the table 3 are copied by the article Eur J Med Chem, 2014, 78, 217-24.
b. the table 8 is same as the published article Bioorg Med Chem. 2015, 23, 498 3674-3680.
c. The table 1 is the same as the published article Eur J Med Chem. 2009, 44, 891-900.
d. The table 4 is the same as published in the article Bioorganic & Medicinal Chemistry, 2014, 22, 2643-2647, etc
d. The sentence in the lines 244- 249 are the same with the published Bioorganic & Medicinal Chemistry, 22, 2014, 2643-2647
Moreover, the authors are not given the meaning of the abbreviations when they are appeared for the first time.
There are many grammatical and syntax errors
a. This review focus on..
b." Thanks to this research …."
c. These natural compounds with wide antibiotic and antitumor activities…
d. " ..further confirmed by total synthesis"
e. ".. Based on these observations we have proposed"
f".." We have to underline, that.."
g." In the group of simple synthesized pyrazolo[4,3-e][1,2,4]triazines [8,9,15,16,19-21] only a few derivatives showed moderate activity against variety of human tumor cell lines." In which tumor cell lines?
It is not clear why the authors give two IC50 values of the same compound (for example tables 4, 8)
Author Response
In the following we discuss the points raised by the Reviewer and state where we have adapted the manuscript
- The manuscript cannot be accepted for publication in the journal of Molecules since the tables and the figures are copied by other already published articles. For example
the figures 2-3 and the table 3 are copied by the article Eur J Med Chem, 2014, 78, 217-24.
The presentation of Figure 2 and 3 is necessary to understand the molecular mechanism of action of novel pyrazolo[4,3-e][1,2,4]triazine sulfonamide derivatives (9a-l). The NH group between the heterocyclic system and the aryl ring in the new series of sulfonamides is a key part of the molecule and it is involved in the interaction with the active site of the protein kinase Abl and CDK2 by forming hydrogen bonds with M318 and L83, respectively. Hence utilizing figures from previous publication enables to visualize this bonds.
According to the suggestion the Table 3 was modified.
Table 3. In vitro antiproliferative activity of new sulfonamide derivatives of pyrazolo[4,3‑e][1,2,4]triazines.
|
Compd. |
R |
MTT assay, IC50 (µM) |
|||
|
K562 |
BV173 |
HL60 |
CCRF-CEM |
||
|
9a |
66±5 |
40 ± 8 |
49±2 |
36±2 |
|
|
9b |
90±8 |
58±5 |
39±1 |
69±8 |
|
|
9c |
27±4 |
22±6 |
55±2 |
20±2 |
|
|
9d |
100±4 |
41±10 |
42±5 |
49±8 |
|
|
9e |
21±5 |
22±4 |
38±1 |
36±12 |
|
|
9f |
102±1 |
47±14 |
56± 6 |
30±2 |
|
|
9g |
98±2 |
36±9 |
24±2 |
30±2 |
|
|
9h |
77±7 |
39±8 |
42±6 |
56±2 |
|
|
9i |
106±8 |
45±11 |
58±3 |
50±1 |
|
|
9j |
>200 |
58±9 |
40±2 |
54±8 |
|
|
9k |
96±3 |
39±8 |
41±1 |
64±6 |
|
|
9l |
101±2 |
42±9 |
44±5 |
57±3 |
|
|
chlorambucil |
|
84±6 |
34±8 |
38±2 |
21±8 |
|
imatinib |
|
13±2 |
20±6 |
55±7 |
45±1 |
- The table 8 is same as the published article Bioorg Med Chem. 2015, 23, 498 3674-3680.
According to the Reviewer suggestion the Table 8 was modified.
Table 8. Cytotoxic and cytostatic activities of new sulfonamides derivatives of pyrazolo[4,3‑e][1,2,4]triazines 24a-k.
|
Compd. |
R1 |
R2 |
R3 |
MTT assay, IC50 (µM) |
[3H]thymidine incorporation, IC50 (µM) |
||
|
MCF-7 |
MDA-MB-231 |
MCF-7 |
MDA-MB-231 |
||||
|
24a |
H |
4-SO2 |
4-methylpiperazin-1-yl |
>200 |
>200 |
>200 |
>200 |
|
24b |
4-CH3 |
H |
NH2 |
154±2 |
171±2 |
155±2 |
148±2 |
|
24c |
4-CH3 |
H |
4-methylpiperazin-1-yl |
>200 |
>200 |
>200 |
>200 |
|
24d |
4-CH3 |
H |
NHCH2CH2OH |
126±2 |
147±2 |
132±2 |
136±2 |
|
24e |
4-CH3 |
H |
(S)-(+)-NHCH2CH(OH)CH3 |
>200 |
>200 |
>200 |
>200 |
|
24f |
4-CH3 |
H |
(R)-(+)-NHCH2CH(OH)CH3 |
>200 |
>200 |
>200 |
>200 |
|
24g |
4-CH3 |
H |
(S)-(+)-NHCH(CH3)CH2OH |
>200 |
>200 |
>200 |
>200 |
|
24h |
4-CH3 |
H |
(R)-(+)-NHCH(CH3)CH2OH |
>200 |
>200 |
>200 |
>200 |
|
24i |
3-Cl |
4-SO2-N-methylpiperazine |
4-methylpiperazin-1-yl |
160±2 |
185±2 |
173±2 |
169±2 |
|
24j |
4-Cl |
H |
4-methylpiperazin-1-yl |
>200 |
>200 |
>200 |
>200 |
|
24k |
4-Cl |
H |
piperazin-1-yl |
>200 |
>200 |
>200 |
>200 |
|
chlorambucil |
|
|
|
97±2 |
93±2 |
56±2 |
49±2 |
- The Table 1 is the same as the published article Eur J Med Chem. 2009, 44, 891-900.
In the Table 1 we have presented only selected structures of the compounds from the original article with their anticancer activity in five human cancer cell lines. However we have replaced Table 1 by the new one.
Table 1. Cytotoxic activity of 3,7-diaryl-5-(3,4,5-trimethoxyphenyl)pyrazolo[4,3-e][1,2,4]-triazines.
|
|
|||||||
|
Compd. |
R1 |
R2 |
MTT assay, IC50 (µM) 1 |
||||
|
CEM |
CEM DNR Bulk |
K562 |
K562 tax |
A549 |
|||
|
1b |
Ph |
2-OCH3-Ph |
109 |
118 |
182 |
94.3 |
5.41 |
|
1c |
Ph |
3-OCH3-Ph |
8.41 |
99.3 |
117 |
159 |
0.61 |
|
1d |
Ph |
4-OCH3-Ph |
117 |
100 |
84.4 |
150 |
39.0 |
|
1e |
Ph |
3,4,5-(OCH3)3-Ph |
65.2 |
129 |
52.6 |
99.9 |
2.16 |
|
2b |
4-OCH3-Ph |
2-OCH3-Ph |
105 |
119 |
178 |
155 |
6.58 |
|
2c |
4-OCH3-Ph |
3-OCH3-Ph |
54.5 |
184 |
115 |
174 |
124 |
|
3b |
3,4,5-(OCH3)3-Ph |
2-OCH3-Ph |
126 |
125 |
182 |
156 |
15.0 |
|
3c |
3,4,5-(OCH3)3-Ph |
3-OCH3-Ph |
48.3 |
109 |
164 |
141 |
163 |
|
3d |
3,4,5-(OCH3)3-Ph |
4-OCH3-Ph |
36.9 |
118 |
66.2 |
161 |
75.6 |
|
3f |
3,4,5-(OCH3)3-Ph |
4-Cl-Ph |
82.2 |
115 |
158 |
196 |
8.26 |
|
4a |
4-Cl-Ph |
Ph |
11.6 |
113 |
220 |
225 |
130 |
|
4b |
4-Cl-Ph |
2-OCH3-Ph |
53.8 |
35.7 |
48.7 |
20.7 |
3.62 |
|
4c |
4-Cl-Ph |
3-OCH3-Ph |
54.7 |
108 |
192 |
219 |
2.86 |
|
4e |
4-Cl-Ph |
3,4,5-(OCH3)3-Ph |
99.9 |
117 |
192 |
210 |
15.0 |
|
4f |
4-Cl-Ph |
4-Cl-Ph |
68.9 |
126 |
190 |
222 |
155 |
1 All experiments were performed as described in the literature [12].
- The table 4 is the same as published in the article Bioorganic & Medicinal Chemistry, 2014, 22, 2643-2647, etc
According to the Reviewer the Table 4 was modified.
Table 4. Cytotoxic and cytostatic activities of new sulfonamides derivatives of pyrazolo[4,3-e]-[1,2,4]triazines 9a-l.
|
Compd. |
R |
MTT assay, IC50 (µM) |
[3H]thymidine incorporation, IC50 (µM) |
||
|
MCF-7 |
MDA-MB-231 |
MCF-7 |
MDA-MB-231 |
||
|
9a |
102±2 |
99±2 |
87±2 |
80±2 |
|
|
9b |
>200 |
>200 |
>200 |
>200 |
|
|
9c |
150±2 |
130±2 |
170±2 |
103±2 |
|
|
9d |
>200 |
>200 |
>200 |
>200 |
|
|
9e |
140±3 |
155±2 |
123±1 |
150±2 |
|
|
9f |
>200 |
>200 |
>200 |
>200 |
|
|
9g |
200±2 |
140±1 |
150±2 |
135±1 |
|
|
9h |
126±1 |
120±1 |
85±1 |
90±1 |
|
|
9i |
>200 |
>200 |
nt 1 |
nt 1 |
|
|
9j |
>200 |
>200 |
nt 1 |
nt 1 |
|
|
9k |
146±1 |
125±2 |
99±1 |
120±2 |
|
|
9l |
200±2 |
140±2 |
nt 1 |
nt 1 |
|
|
chlorambucil |
|
97±2 |
93±2 |
56±2 |
49±2 |
1 nt – not tested.
- The sentence in the lines 244- 249 are the same with the published Bioorganic & Medicinal Chemistry, 22, 2014, 2643-2647
According to the Reviewer suggestion the fragment has been redrafted.
- Moreover, the authors are not given the meaning of the abbreviations when they are appeared for the first time.
We have clarified the definition of missing abbreviations in the “ Abbreviation section”
- " In the group of simple synthesized pyrazolo[4,3-e][1,2,4]triazines [8,9,15,16,19-21] only a few derivatives showed moderate activity against variety of human tumor cell lines." In which tumor cell lines?
We agree that ‘variety of human tumour cell lines” is not explained as fully as necessary. According to the Reviewer's suggestion the type of cancer cells were listed in the text.
- It is not clear why the authors give two IC50 values of the same compound (for example tables 4, 8)
We agree it would be useful to briefly explain two IC50 values were given. According to the suggestion we have given the information about two tests which are used to assess the cytotoxic and cytostatic activity of novel synthesized compounds. (“Moreover, for both sulfonamide groups the potential anticancer activity in MCF-7 and MDA-MB-231 cells was determined by [3H]thymidine incorporation assay and MTT test, where proliferation and viability of breast cancer cells were analyzed (Table 4 and 5) [44]. All tested compounds showed concentration dependent activity but with different potency.” )
We want to thank the Reviewer for critical but constructive comments. We apologize for grammatical and syntax errors. We hope the corresponding revisions have strengthened the paper in multiple ways.

Round 2
Reviewer 1 Report
Comments and Suggestions for Authors:
The manuscript has been improved and has now much better quality, but the manuscript still need some supplement and correction before it should be considered for publication.
There are some notes, as follows:
- Lines 21, 376-378, 407, 413, 417, 430 and 431: The nomenclature of pyrazolo[4,3-e][1,2,4]triazolo[4,5-b][1,2,4]triazines should be corrected as pyrazolo[4,3-e][1,2,4]triazolo[4,3-b][1,2,4]triazines and pyrazolo[4,3-e]tetrazolo[4,5-b][1,2,4]triazine should be corrected as pyrazolo[4,3-e]tetrazolo[1,5-b][1,2,4]triazine.
- In Introduction section: It should statement detailing the aim, scope, and relevance of the topic that to be reviewed. Beside, an outline should be described. It is useful to potential readers of the review.
- Line 139: 1-phenyl-3-methyl-5-anilinopyrazolo[4,3-e][1,2,4]triazine should be corrected as 3-methyl-1-phenyl-5-phenylaminopyrazolo[4,3-e][1,2,4]triazine
The organization is most important in the review paper. The correct nomenclature of a structure is a kind of organization that benefit forever.
Which nomenclature of the following is better for heterocyclic compound 1?
(a) 1H-pyrazolo[4,3-e][1,2,4]triazine (b) 1H-pyrazolo[4,5-e][1,2,4]triazine
Which nomenclature of the following is better for heterocyclic compound 2?
(a) tetrazolo[1,5-b][1,2,4]triazine (b) tetrazolo[4,5-b][1,2,4]triazine
Which nomenclature of the following is much better for heterocyclic compound 3?
(a) pyrazolo[4,3-e]tetrazolo[1,5-b][1,2,4]triazine
(b) pyrazolo[4,3-e]tetrazolo[4,5-b][1,2,4]triazine
(c) pyrazolo[4,5-e]tetrazolo[1,5-b][1,2,4]triazine
(d) pyrazolo[4,5-e]tetrazolo[4,5-b][1,2,4]triazine
Author Response
Respond Review 1
The responses to the comments are listed below point by point.
- Lines 21, 376-378, 407, 413, 417, 430 and 431: The nomenclature of pyrazolo[4,3-e][1,2,4]triazolo[4,5-b][1,2,4]triazines should be corrected as pyrazolo[4,3-e][1,2,4]triazolo[4,3-b][1,2,4]triazines and pyrazolo[4,3-e]tetrazolo[4,5-b][1,2,4]triazine should be corrected as pyrazolo[4,3-e]tetrazolo[1,5-b][1,2,4]triazine.
Nomenclature was corrected according to reviewer’s comments.
- In Introduction section: It should statement detailing the aim, scope, and relevance of the topic that to be reviewed. Beside, an outline should be described. It is useful to potential readers of the review.
The following text was added to the introduction:
This review presents the methods for the synthesis and functionalization of the pyrazolo[4,3-e][1,2,4]triazine ring system, which in the group of fused pyrazolotriazines is little known and also little described in the literature. It is known that proper functionalization of the heterocyclic core is a key element to design new molecules with potential biological properties. Therefore, the collection of the results of the current scientific research on this valuable heterocyclic system is necessary and justified. The results of experimental studies published so far have shown that a series of derivatives of this heterocyclic system possess various biological activity including antitumor property, which is the main and leading topic of this review. The paper also presents the methods used for the preparation of pyrazolo[4,3-e][1,2,4]triazine sulfonamide derivatives, their activity against cancer cell lines and the inhibition of two carbonic anhydrase isozymes (CAIX and CAXII) that are highly overexpresed in hypoxic tumors and show very restricted expression in normal tissues. Among the presented data, the tricyclic pyrazolo[4,3-e][1,2,4[triazines fused with triazole or tetrazole ring are of particular interest. They constitute new groups of heterocyclic systems and are characterized by high antitumor activity and it seems that they may be a source of new chemotherapeutic agents.
- Line 139: 1-phenyl-3-methyl-5-anilinopyrazolo[4,3-e][1,2,4]triazine should be corrected as 3-methyl-1-phenyl-5-phenylaminopyrazolo[4,3-e][1,2,4]triazine
It was corrected/
- The organization is most important in the review paper. The correct nomenclature of a structure is a kind of organization that benefit forever.
We have revised our manuscript again and now believe that the manuscript is acceptable for publication in Molecules.
Reviewer 3 Report
The manuscript can be accepted for publication in Molecules, after major revision. The authors should revise the manuscript according to the following comments.The authors should take permission for the figures 2 and 3 from the published article Eur J Med Chem, 2014, 78, 217-24
Author Response
Respond Review 3
The responses to the comments are listed below.
The manuscript can be accepted for publication in Molecules, after major revision. The authors should revise the manuscript according to the following comments. The authors should take permission for the figures 2 and 3 from the published article Eur J Med Chem, 2014, 78, 217-24
Figures 2 and 3 have been deleted. We have not been able to contact all the co-authors of the original article (Eur J Med Chem, 2014, 78, 217-24) and the figures have been removed due to the limited time needed for correction.